# Bafilomycin A1 Molecular Effect on ATPase Activity of Subcellular Fraction of Human Colorectal Cancer and Rat Liver

**DOI:** 10.3390/ijms25031657

**Published:** 2024-01-29

**Authors:** Solomiia Bychkova, Mykola Bychkov, Dani Dordevic, Monika Vítězová, Simon K.-M. R. Rittmann, Ivan Kushkevych

**Affiliations:** 1Department of Human and Animal Physiology, Faculty of Biology, Ivan Franko National University of Lviv, 79005 Lviv, Ukraine; solomiya.bychkova@lnu.edu.ua; 2Department of Therapy No. 1, Medical Diagnostic and Hematology and Transfusiology of Faculty of Postgraduate Education, Danylo Halytsky Lviv National Medical University, 79010 Lviv, Ukraine; mbychkov21@gmail.com; 3Department of Plant Origin Food Sciences, Faculty of Veterinary Hygiene and Ecology, University of Veterinary Sciences Brno, 612 42 Brno, Czech Republic; dordevicd@vfu.cz; 4Department of Experimental Biology, Faculty of Science, Masaryk University, 625 00 Brno, Czech Republic; vitezova@sci.muni.cz; 5Department of Functional and Evolutionary Ecology, Archaea Physiology & Biotechnology Group, Universität Wien, 1030 Wien, Austria

**Keywords:** molecular mechanisms, colon cancer, ATPase, autophagy, hepatocytes, liver, NAADP, biomarkers, bafilomycin A1, Ca^2+^ store

## Abstract

Bafilomycin A1 inhibits V-type H^+^ ATPases on the molecular level, which acidifies endo-lysosomes. The main objective of the study was to assess the effect of bafilomycin A1 on Ca^2+^ content, NAADP-induced Ca^2+^ release, and ATPase activity in rat hepatocytes and human colon cancer samples. Chlortetracycline (CTC) was used for a quantitative measure of stored calcium in permeabilized rat hepatocytes. ATPase activity was determined by orthophosphate content released after ATP hydrolysis in subcellular post-mitochondrial fraction obtained from rat liver as well as from patients’ samples of colon mucosa and colorectal cancer samples. In rat hepatocytes, bafilomycin A1 decreased stored Ca^2+^ and prevented the effect of NAADP on stored Ca^2+^. This effect was dependent on EGTA–Ca^2+^ buffers in the medium. Bafilomycin A1 significantly increased the activity of Ca^2+^ ATPases of endoplasmic reticulum (EPR), but not plasma membrane (PM) Ca^2+^ ATPases in rat liver. Bafilomycin A1 also prevented the effect of NAADP on these pumps. In addition, bafilomycin A1 reduced Na^+^/K^+^ ATPase activity and increased basal Mg^2+^ ATPase activity in the subcellular fraction of rat liver. Concomitant administration of bafilomycin A1 and NAADP enhanced these effects. Bafilomycin A1 increased the activity of the Ca^2+^ ATPase of EPR in the subcellular fraction of normal human colon mucosa and also in colon cancer tissue samples. In contrast, it decreased Ca^2+^ ATPase PM activity in samples of normal human colon mucosa and caused no changes in colon cancer. Bafilomycin A1 decreased Na^+^/K^+^ ATPase activity and increased basal Mg^2+^ ATPase activity in normal colon mucosa samples and in human colon cancer samples. It can be concluded that bafilomycin A1 targets NAADP-sensitive acidic Ca^2+^ stores, effectively modulates ATPase activity, and assumes the link between acidic stores and EPR. Bafilomycin A1 may be useful for cancer therapy.

## 1. Introduction

A class of macrolide antibiotics known as bafilomycins is created from several different streptomycetes. These compounds’ chemical structure is determined by a 16-membered lactone ring scaffold. A variety of biological activities, including anti-parasitic, anti-tumor, immunosuppressive, and anti-fungal action, are connected with bafilomycins [1]. Bafilomycin A1 is widely used as an autophagy inhibitor in various tissues [2,3,4]. In addition, bafilomycin A1 is also used as a specific inhibitor of V-type H^+^ ATPase [5,6], which pumps protons into the lumen of organelles such as lysosomes [7]. It is well known that endosomes and lysosomes are important membrane-bound organelles that are essential for the normal functioning of the eukaryotic cell. Interaction disruption between endosomes and lysosomes may contribute to cancer development [8].

By preventing V-ATPase-dependent acidification and autophagosome-lysosome fusion, bafilomycin A1 has been demonstrated to interfere with autophagic flux [9]. Because autophagy involves the recycling of organelles and lysosomal breakdown, it is thought to be a crucial survival strategy for both cancerous and healthy cells. Research has demonstrated that bafilomycin A1-induced macroautophagy suppression reduces colon cancer cell proliferation and triggers apoptosis [10], suppresses the growth of HCC cells [11], inhibits autophagy flux in diffuse large B-cell lymphoma [12], and inhibits respiration in mitochondria of Nemeth-Kellner lymphoma [13]. Thus, bafilomycin A1 could be an effective therapeutic agent in cancer therapy [14]. Furthermore, bafilomycin A1 inhibited autophagosome–lysosome fusion and acidification, which resulted in a marked rise in cytosolic calcium concentration [9]. These acidic organelles are known to contain high amounts of Ca^2+^ [15]. Faris and others demonstrated that nicotinic acid adenine dinucleotide phosphate (NAADP) induces intracellular Ca^2+^ release in primary cultures of metastatic colorectal cancer Ca^2+^ imaging and molecular biology techniques [16]. The ATPase activity of the subcellular fraction of human colon cancer samples has recently been revealed to be impacted by NAADP, which helps to stimulate Ca^2+^ release from acidic organelles [17]. Bafilomycin A1 has been shown to inhibit NAADP-induced Ca^2+^ release from these organelles in rat hepatocytes [18], though it is unknown how bafilomycin A1 affects ATPase activity in cancer tissues.

Ca^2+^ ATPases are in charge of creating sharp Ca^2+^ gradients across intracellular membranes or the plasma membrane and preserving a low baseline Ca^2+^ level in the cytoplasm. [19], and remodeling Ca^2+^ signaling is an important step in cancer progression [19,20,21]. The growth of cancer is facilitated by changes in colon cancer cells’ expression of the plasma Ca^2+^ pump (PMCA) and the corresponding modifications in the calcium released by the cell [21,22]. Differentiation of HT-29 colon cancer cells is associated with the upregulation of PMCA4, but no significant change in PMCA1 [22]. It was found that the simultaneous presence of bafilomycin A1 and NAADP completely inhibits PMCA activity in mouse NK/Ly cells [23], but the effect of bafilomycin A1 on PMCA activity in human colon cancer has not been studied.

Maintenance of high endoplasmic reticulum calcium concentration through the action of sarco/endoplasmic reticulum calcium ATPases (SERCAs) is critical for many cellular functions involved in intracellular signaling, control of proliferation, programmed cell death, or synthesis of mature proteins [24]. Increased expression of SERCA 2 was found in human colorectal cancer and was associated with advanced tumor stage and tumorigenesis [25]. Meanwhile, in several other cancers, SERCA3 expression is selectively downregulated [24]. We found that NAADP decreased the activity of SERCA and basal Mg^2+^ ATPase in the post-mitochondrial fraction of mouse lymphomas, and bafilomycin A1 prevented the effects of NAADP on the activity of these pumps [13]. This confirmed the function of these pumps in the context of bafilomycin-sensitive acid stores. Recently, it was found that NAADP causes a decrease in Ca^2+^ ATPase EPR activities and an increase in basal ATPase activity in human colon cancer samples [17], but the effect of bafilomycin A1 on the activity of these pumps has not been studied. It is known that Na^+^/K^+^ ATPases are crucial for cancer cell adhesion, motility, and migration [20,26]. Inhibitors of Na^+^/K^+^ ATPase (ouabain and digoxin) showed anti-tumor effects on multicellular tumor spheroids of hepatocellular carcinoma [27], and another inhibitor (gentiopicrin) exerted anticancer activity on human colon cancer [28]. Such glycosides as gentiopicrin are known to be used in traditional medicine for the treatment of heart disease, as they selectively inhibit Na^+^/K^+^ ATPase and increase intracellular Ca^2+^ concentration.

NAADP decreased Na^+^/K^+^ ATPase activity in mouse NK/Ly cells [13], and the simultaneous presence of bafilomycin A1 and NAADP caused a stronger inhibition of Na^+^/K^+^ ATPase activity and, at the same time, a strong decrease in molecular oxygen consumption in mitochondria and uncoupling of oxidative phosphorylation and respiration was observed in mouse NK/Ly cells [13]. Moreover, there was a dramatic decrease in Na^+^/K^+^ ATPase activity by application of NAADP, which could be explained by the release of Ca^2+^ from acid stores by NAADP in human colon cancer tissue samples [17]. Thus, attacking acid stores is an effective tool in cancer therapy. The effect of bafilomycin A1 on Na^+^/K^+^ ATPase activities in human colon cancer remains unexplored.

Therefore, the primary goal of the investigation was to determine the functional connections between acid Ca^2+^ storage (endo-lysosomes), extracellular phosphatase (EPR), and other active ion transport systems (ATPases) by examining the impact of bafilomycin A1 on Ca^2+^ content, NAADP-induced Ca^2+^ release, and ATPase activity in rat hepatocytes. The study also included an investigation into the impact of bafilomycin A1 on ATPase activity in human colon cancer samples to evaluate its potential use as a pharmacological agent in cancer therapy.

## 2. Results

### 2.1. Bafilomycin A1 Effect on ATPase Activity in Subcellular Fraction of Human Colon Mucosal Tissue Samples and Colorectal Cancer Tissue Samples

It was found that the average Na^+^/K^+^ ATPase activity in the human colon mucosa samples was 4.00 ± 0.61 μmol P_i_/mg protein per h. After the addition of bafilomycin A1 (0.001 mM) to the incubation solution of the post-mitochondrial subcellular fraction, this index decreased to 2.43 ± 0.23 μmol P_i_/mg protein per h. Consequently, 1.6 times less Na^+^/K^+^ ATPase activity was seen in the subcellular portion of human colon mucosal tissue when bafilomycin A1 (0.001 mM) was added (*n* = 10; *p* ≤ 0.05) (Figure 1).

In the subcellular fraction of human colon cancer samples, the average Na^+^/K^+^ ATPase activity was 2.37 ± 0.34 μmol P_i_/mg protein per h. Bafilomycin A1 (0.001 mM) caused the statistically significant (*n* = 10; *p* ≤ 0.05) decrease in this index to 0.94 ± 0.32 μmol P_i_/mg protein per h. Thus, bafilomycin A1 (0.001 mM) decreased Na^+^/K^+^ ATPase activity in the subcellular fraction of human colorectal cancer samples by 2.52 folds. This effect is more pronounced than in samples of healthy mucosa.

The median gene expression of Na^+^/K^+^-ATPase subunits (Figure 2) of colon tumor vs. normal samples for Na^+^/K^+^ ATPase (ATP1A1)—2.88-fold increase in cancer (535.19 cancer/186.04 normal); Na^+^/K^+^ ATPase (ATP1B1)—3.47-fold increase in cancer (235.35 cancer/67.88 normal); Na^+^/K^+^ ATPase (ATP1B3)—1.46-fold increase in cancer (174.84 cancer/119.05 normal) according to Gepia (http://gepia.cancer-pku.cn/detail.php?gene, accessed on 9 January 2024).

In the human colon mucosa samples, the values of endoplasmic reticulum Ca^2+^ ATPase of EPR activity were 1.95 ± 0.30 μmol P_i_/mg protein per h (Figure 1). After the addition of bafilomycin A1 (0.001 mM), the activity of EPR Ca^2+^ ATPase increased to 2.59 ± 0.24 μmol P_i_/mg protein per h. Thus, bafilomycin A1 increased the activity of Ca^2+^ ATPase of EPR by 1.32-fold in the colon mucosa tissue samples (*n* = 10; *p* ≤ 0.05). In human colon cancer samples, Ca^2+^ ATPase EPR activity was 1.42 ± 0.45 μM μmol P_i_/mg protein per h (Figure 1). After the addition of bafilomycin A1 (0.001 mM) to the incubation medium of the cancer tissue samples, the average value of ATPase activity was 3.59 ± 0.34 μmol P_i_/mg protein per h. Thus, bafilomycin A1 (0.001 mM) statistically (*n* = 10; *p* ≤ 0.05) increased the activity of Ca^2+^ ATPase EPR in colon cancer tissue samples by 2.58-fold. In human colon mucosal tissue samples, plasmatic membrane (PM) Ca^2+^ ATPase activity was 6.25 ± 0.56 μmol P_i_/mg protein per h. Under the influence of bafilomycin A1, the activity of the Ca^2+^ ATPase of the PM was 4.44 ± 0.28 μmol P_i_/mg protein per h. Thus, bafilomycin A1 (0.001 mM) decreased PM Ca^2+^ ATPase activity 1.40-fold in human intestinal mucosal tissue samples (*n* = 10; *p* ≤ 0.05). In human colon cancer samples, PM Ca^2+^ ATPase activity averaged 1.79 ± 0.26 μmol P_i_/mg protein per h. After the addition of bafilomycin A1, the average Ca^2+^ ATPase activity of PM was 1.66 ± 0.28 μmol P_i_/mg protein per h. Thus, bafilomycin A1 caused no changes in the Ca^2+^ ATPase activity of PM in the cancer tissue samples. In the colon mucosa tissue samples, the basal ATPase activity was 6.41 ± 1.03 μmol P_i_/mg protein per h.

When bafilomycin (0.001 mM) was added, the basal ATPase activity increased to 14.78 ± 6.21 μM μmol P_i_/mg protein per h. Thus, bafilomycin A1 increased basal ATPase activity 2.30-fold in human intestinal mucosal tissue samples (*n* = 10; *p* ≤ 0.05). In colon cancer tissue samples, the indicators of basal ATPase activity varied from 2.10 to 25.63 and averaged 12.56 ± 3.11 μmol P_i_/mg protein per h (see Figure 1). When bafilomycin A1 (0.001 mM) was added to the cancer samples, the indicators of ATPase activity ranged from 3.34 to 35.72, with an average value of 18.60 ± 6.29 μmol P_i_/mg protein per h. Thus, the results are underlying the statistically 1.48-fold increase in basal ATPase activity of human colon cancer tissue samples by bafilomycin A1 (*n* = 10; *p* ≤ 0.05).

### 2.2. Modulation of ATPase Activity by Bafilomycin A1 and Its Impact on NAADP-Induced Effects in Subcellular Rat Liver Fractions

To investigate the effect of bafilomycin A1 (0.001 mM) on ATPase activity, the post-mitochondrial subcellular fraction of rat liver obtained by the differential centrifugation method was used. The results indicate that the mean activity of Na^+^/K^+^ ATPase of the post-mitochondrial fraction in rat liver was 3.20 ± 0.24 μmol P_i_/mg protein per h (*n* = 6) in the control (Figure 3).

Bafilomycin A 1 (0.001 mM) addition to the incubation medium of the post-mitochondrial fraction of rat liver resulted in a decrease in the mean activity of Na^+^/K^+^ ATPase to 1.95 ± 0.98 μmol P_i_/mg protein per h (*n* = 6). Thus, bafilomycin A 1 (0.001 mM) reduced Na^+^/K^+^ ATPase activity by 41.67% (*p* > 0.05) in the post-mitochondrial fraction of rat liver. Bafilomycin A1 (0.001 mM) significantly increased the activity of Ca^2+^ ATPases of EPR in the subcellular fraction of rat liver from 3.03 ± 0.93 to 8.09 ± 4.47 μmol P_i_/mg protein per h. This was an approximately three-fold increase in Ca^2+^ pump activity of EPR (*p* ≤ 0.05, *n* = 6). In contrast, bafilomycin A1 (0.001 mM) did not statistically alter PM Ca^2+^ ATPase activity of the post-mitochondrial fraction in rat liver, but it increased basal Mg^2+^ ATPase activity more than two-fold from 12.26 ± 1.29 to 20.33 ± 2.50 μmol P_i_/mg protein per h (*p* ≤ 0.05, *n* = 6) (Figure 3).

Simultaneous addition of bafilomycin A1 (0.001 mM) and NAADP (7 μM) to the incubation medium caused a greater decrease in the activity of Na^+^/K^+^ pumps of the rat liver subcellular fraction compared with the control (0.33 ± 0.29 μmol Pi/mg protein per h) (*p* ≤ 0.05, *n* = 6). NAADP and bafilomycin A1 caused a greater increase in basal ATPase activity to 30.24 ± 2.32 μmol P_i_/mg protein per h, statistically significant compared with both the medium containing NAADP alone and bafilomycin A1 alone. The activity of Ca^2+^ ATPases of PM in the simultaneous presence of bafilomycin A1 and NAADP in the incubation medium was 3.25 ± 1.63 μmol P_i_/mg protein per h. This was statistically significant (*p* ≤ 0.05) compared with the medium containing bafilomycin A1 alone and much lower compared with the medium containing NAADP. The activity of the Ca^2+^ ATPases EPR of the rat liver subcellular fraction was 6.85 ± 2.57 μmol P_i_/mg protein per h when bafilomycin A1 and NAADP were added simultaneously to the incubation medium (see Figure 3). This was not statistically authentic to the medium with NAADP or to the medium with bafilomycin A1 alone. Thus, bafilomycin A1 also prevented the effect of NAADP on the activity of the Ca^2+^ ATPases of EPR in rat liver.

### 2.3. Bafilomycin Affected Stored Calcium Content and Prevented NAADP-Induced Changes of Stored Calcium in Rat Hepatocytes

In the first phase of our investigation, we assessed the effects of bafilomycin in a medium without EGTA–Ca^2+^ buffer on the calcium content in permeabilized rat hepatocytes as well as the simultaneous effects of bafilomycin (20, 0.001 mM) and NAADP on the calcium content in the permeabilized rat (Figure 4).

To find out the effect of bafilomycin A1 on endo-lysosomal stores, its effect on Ca^2+^ content in permeabilized rat hepatocytes was investigated. Bafilomycin A1 at different concentrations (20; 0.04; 0.001 mM) was used to study its effect on stored calcium in permeabilized rat hepatocytes incubated in a medium without EGTA–Ca^2+^ buffer. We observed that bafilomycin A1 significantly decreased CTC–Ca^2+^ chemiluminescence in the millimolar concentration range (20 and 0.04 mM), corresponding to the decrease in the content of Ca^2+^ in intracellular organelles of permeabilized rat hepatocytes. The amount of calcium that was stored remained unchanged despite the reduced bafilomycin A1 concentration (0.001 mM) (Figure 4A).

To investigate the relationships between NAADP-sensitive stores and bafilomycin-sensitive ones, we used these two drugs in the incubation medium. No changes in stored calcium were observed in permeabilized hepatocytes incubated in the medium with simultaneous presence of bafilomycin A1 (20 mM) and NAADP (7 μM) compared with the medium with only bafilomycin A1 (Figure 4B). The simultaneous effect of a lower concentration of Bafilomycin A1 (0.001 mM) and NAADP (7 μM) resulted in a statistically significant increase in stored calcium by 40.23 ± 3.47% (*p* ≥ 0.001) compared to the medium with bafilomycin A1 (0.001 mM) alone. These series of experiments were performed in an incubation solution without chelating agents. Previously, we found that NAADP releases Ca^2+^ in permeabilized rat hepatocytes and that the NAADP-induced changes in Ca^2+^ storage in these cells depend on the concentration of EGTA–Ca^2+^ buffer in the cell incubation medium.

A series of experiments were conducted using EGTA–Ca^2+^ buffers to change the concentration of free calcium in the incubation medium in order to control the amount of calcium present better (Figure 5).

Ca^2+^ salts (0.050 mM) and EGTA (0.100 mM) were present in high amounts in medium A, while EGTA (0.05 mM) and Ca^2+^ salts (0.025 mM) were present in lower concentrations in medium B. The EGTA–Ca^2+^ buffer solution kept the concentration of free calcium constant in both situations at 240 nM, which is the hepatocytes’ physiological resting state.

In any solution containing Ca^2+^ chelators, bafilomycin A1 (0.001 mM) did not affect the amount of calcium that was stored in rat hepatocytes (see Figure 4). The impact of NAADP was influenced by the concentration of EGTA, as we have found in our previous work [18]. In medium B with low EGTA content as well as in medium without chelators, NAADP (7 μM) led to a reduction in calcium that was stored. When bafilomycin A1 (0.001 mM) and NAADP (7 μM) were mixed, we discovered that the calcium content increased by 41.36 ± 3.92% (*p* ≥ 0.001) compared to NAADP alone (Figure 5B). However, this effect was not statistically significant against either the control or bafilomycin A1. Consequently, it was found that bafilomycin A1 (0.001 mM) blocked NAADP’s effects across all medium types.

## 3. Discussion

Previously, we found that NAADP affects ATPase activity in cancer samples, probably due to Ca^2+^ release from acidic stores, as we had suspected [17]. In this article, we investigated the effect of bafilomycin A1 on the ATPase activity of human colon cancer tissue samples. It is well known that differentiation of colorectal cancer cells is associated with altered Ca^2+^ homeostasis and expression of specific sarcoplasmic/endoplasmic reticulum calcium ATPase (ERCA) isoforms [21]. Changes in SERCA expression and activity cause cellular cancer, induce ER stress, and trigger apoptosis linked to ER stress. [29]. In addition to the Ca^2+^ ATPase of the EPR (SERCA), the important role of the Ca^2+^ ATPase of the PM (PMCA) has also been demonstrated, playing an important role in remodeling Ca^2+^ homeostasis in human colon cancer cells [22].

We observed a three-fold higher activity of Ca^2+^ ATPase of PM compared with the activity of Ca^2+^ ATPase of EPR in normal human colon mucosa (6.25 ± 0.56 versus 1.95 ± 0.30 μmol P_i_/mg protein per h) (Figure 1). However, in colon cancer tissue samples, the activity of these two pumps was low (1.79 ± 0.26 μmol P_i_/mg protein per h for PMCA). At the same time, the activity of the Ca^2+^ ATPase of EPR (SERCA) only tended to decrease and remained low in colon cancer tissues (1.42 ± 0.45 μmol P_i_/mg protein per h). It is important to note that the activity of PMCA was reduced more than 3.5-fold in cancer tissue compared to healthy mucosa (Figure 1). Our results are consistent with other studies on colorectal cancer that have shown lower PMCA4 expression in tumors compared to normal tissues [21,30,31], and loss of SERCA3 expression was an early event during colon carcinogenesis [24,32]. Additionally, it has been shown that the expression of the SERCA3 protein is down-regulated in gastric and colorectal cancer cell lines, indicating that cell differentiation in vitro enhances its expression [33]. Moreover, the expression of Ca^2+^ pumps was shown to be highly regulated in breast cancer cells in a subtype-specific manner [34]. Thus, we observed decreased activity of both Ca^2+^ ATPase of EPR and PM in cancer tissues compared to normal human mucosa, which is consistent with protein expression data obtained by other research groups. It was found here that bafilomycin A1 (Figure 1) effectively increased EPR Ca^2+^ ATPase activity in both normal human mucosa and colon cancer tissue samples by 1.32-fold in colon mucosa tissue samples (*n* = 10; *p* ≤ 0.05) and by 2.58-fold (*n* = 10; *p* ≤ 0.05) in colon cancer tissue samples. We hypothesize that this is due to the release of Ca^2+^ from endo-lysosomal stores due to inhibition of the H^+^ pump by bafilomycin A1, resulting in locally high Ca^2+^ concentrations that activate the Ca^2+^ ATPase of the EPR. This is further confirmation of a very close and strong contact site between these different Ca^2+^ pools (lysosomes and EPR), which are still present even in the subcellular fraction under these experimental conditions. A clearly visible membrane contact sites between lysosomes and ER membranes in human cardiac mesenchymal stromal cells were recently confirmed by transmission electron microscopy [16,35], and early similar nanojunctions between lysosomes and sarcoplasmic reticulum were shown in pulmonary artery smooth muscle cells [36]. Therefore, bafilomycin A1 increases the activity of the EPR Ca^2+^ pump due to tight colocalization of EPR membranes with lysosomes. Based on our experimental data, we hypothesized that there are similar contact sites between bafilomycin-sensitive acidic stores and EPR in the subcellular fraction of normal human intestinal mucosa as in colorectal cancer. This assumption is consistent with the observations of other research groups [35,36,37].

We also observed that bafilomycin A1 decreased the activity of Ca^2+^ ATPase of PM by 1.40-fold in human colon mucosa tissue samples (*n* = 10; *p* ≤ 0.05). This may be due to the fact that bafilomycin A1 induces a significantly lower cytosolic pH due to the inhibition of the H^+^ pump of the acidic store. It is known that cytosolic acidification can inhibit PMCA [38], just as PMCA simultaneously affects intracellular Ca^2+^ regulation and pH [39].

It was also shown that in the subcellular fraction of human colorectal tissue samples, bafilomycin A1 did not alter the activity of Ca^2+^ ATPase of PM (Figure 1). We hypothesized that the Ca^2+^ ATPase of PM in cancer cells is not in close colocalization with the endo-lysosomal system, possibly as a consequence of a disturbance in membrane circulation. It was found that the activity of basal Mg^2+^ ATPase was two-fold higher in the subcellular fraction of human colorectal cancer tissue samples than in normal mucosa tissue.

It was observed that bafilomycin A1 increased basal ATPase activity in normal human intestinal mucosal tissue samples as well as in human colon cancer tissue samples. This effect could not be related to the release of Ca^2+^ from acidic stores because basal Mg^2+^ ATPase activity was estimated in solution with EGTA. The effect of bafilomycin A1 on basal Mg^2+^ ATPase activity is realized by a change in pH due to inhibition of the H^+^ pump. It was shown that the pH dependence of the enzymatic activity of basal Mg^2+^ ATPase was not bell-shaped [40] but was characterized by the linearity in the range of values of hydrogen index 6.0–8.0 [41].

It was found that the activity of Na^+^/K^+^ ATPase of the subcellular fraction was higher in the human colon mucosa samples than in the colon cancer samples (4.00 ± 0.61 versus 2.37 ± 0.34 μmol P_i_/mg protein per h). Apparently, this is related to the type of isoform of Na^+^/K^+^ ATPase. It was shown that the alpha-3 isoform of Na^+^/K^+^ ATPase was upregulated in human colon cancer (Figure 2), but the alpha-1 isoform was downregulated [42]. The activity of Na^+^/K^+^ ATPase can lead to the invasion of endocrine-resistant breast cancer cells [43]. High expression of the alpha-1 subunit of Na^+^/K^+^ ATPase has been associated with tumor development and clinical outcomes in gastric cancer [44]. We also observed that bafilomycin A1 decreased Na^+^/K^+^ ATPase activity in human colon mucosal tissue samples and in human colon cancer samples. Importantly, the inhibitory effect of bafilomycin A1 on Na^+^/K^+^ pump activity was more pronounced in cancer tissue than in healthy tissue (see Figure 1). Previously, we observed a similar effect of NAADP on the activity of this protein in colon cancer tissue samples [17]. The simultaneous presence of bafilomycin A1 and NAADP resulted in a stronger inhibition of Na^+^/K^+^ ATPase activity in mouse NK/Ly cells [13]. There is a favorable correlation between the expression levels of the sodium pump-α3 subunit and metastasis in colorectal cancer [45]. Thus, inhibition of Na^+^/K^+^ ATPase could significantly inhibit the migration of colorectal cancer cells. Our results imply that endo-lysosomal agents bafilomycin A1 or NAADP [17] or both [13] are effective in inhibiting Na^+^/K^+^ ATPase activity in various cancers.

The ATPase activity of the rat subcellular fraction under the combined action of bafilomycin A1 was also examined (Figure 3). We found that bafilomycin A1 significantly increased the activity of the Ca^2+^ ATPases of the EPR in the subcellular fraction of rat liver but not that of the Ca^2+^ ATPases of the PM. This observation fully confirmed our assumption about the activation of the Ca^2+^ ATPase of the EPR by bafilomycin A1, as we had suspected above on the basis of experiments on its effect on Ca^2+^ content in rat hepatocytes. Furthermore, because bafilomycin A1 increases the activity of the Ca^2+^ ATPase in the EPR but not the PM, it was postulated that the bafilomycin-responsive acid storage is tightly associated with the EPR but not the PM. This observation is consistent with Ronco V. (2015) [37] that there is a functional Ca^2+^-mediated cross-talk between endolysosomal and endoplasmic reticulum Ca^2+^ pools during NAADP-induced Ca^2+^ signaling.

It was also observed that bafilomycin A1 reduced Na^+^/K^+^ ATPase activity in the post-mitochondrial fraction of rat liver (Figure 3). It is important to note that the subcellular fraction is a heterogeneous mixture of membrane vesicles formed from different membranes: endosomes, lysosomes, EPR, and PM. In addition to Na^+^/K^+^ ATPase in the PM, it is also possible that it is present in the membranes of the endo-lysosomal store [36] as a result of PM invagination. Most likely, the effect of bafilomycin A1 on Na^+^/K^+^ ATPase is realized by a change in pH or/and released Ca^2+^. It is likely that the Ca^2+^ that bafilomycin A1 releases from acidic storage inhibits the function of Na^+^/K^+^ ATPase. Higher intracellular Ca^2+^ has been shown to have this inhibitory impact on the sodium pump in human red blood cells, for instance [46]. Another explanation could be that bafilomycin A1 decreases Na^+^/K^+^ ATPase activity due to a change in pH. It has been shown that acidification to pH values below 6.0 inhibits this protein [41]. In lysosomal membranes of the liver, in addition to the bafilomycin A1-sensitive H^+^ ATPase, the bafilomycin A1-insensitive Mg^2+^ ATPase has also been identified [47]. Bafilomycin A1 was also found to increase the activity of basal Mg^2+^ ATPase by more than two-fold. It is explained that the effect of bafilomycin A1 on the activity of basal Mg^2+^ ATPase is more likely due to a change in pH. The combined action of bafilomycin A1 and NAADP causes an even greater inhibition of Na^+^/K^+^ pump activity and results in an even greater increase in basal ATPase activity. It has already been established that NAADP decreases the activity of basal Mg^2+^ ATPase and the activity of Na^+^/K^+^ ATPase [48].

Therefore, bafilomycin enhances the effect of NAADP on inhibiting the activity of Na^+^/K^+^ pumps. We hypothesize that Na^+^/K^+^ ATPase may be active in endosomal organelles, i.e., bafilomycin and NAADP may reduce pump activity by increasing acidification. It is also possible that there is a NAADP-sensitive but bafilomycin-insensitive store in this intracellular fraction. The basal activity of Mg^2+^ ATPase is determined in the presence of Mg^2+^ and ATP ions at millimolar concentrations and with the addition of EGTA. Thus, the effect of bafilomycin A1 on basal Mg^2+^ ATPase is not associated with an increase in Ca^2+^ concentration due to the presence of high EGTA in the incubation medium of the subcellular fraction. The effect is most likely due to the pH change caused by bafilomycin. It is known that basal Mg^2+^ ATPase activity has a distinct optimum at physiological pH values (7.5–8.5) in the membrane of various tissues and significantly inhibits higher and lower pH values [49,50,51]. Therefore, NAADP, like bafilomycin A1, exerts a unidirectional effect on these systems of active ion transport. Therefore, our assumption above was correct: the effect of bafilomycin A1 on the activity of the Na^+^/K^+^ pump and basal Mg^2+^ ATPase is achieved by a change in pH or Ca^2+^. NAADP was previously found to cause a significant increase in Ca^2+^ ATPase activity of the subcellular fraction of rat liver [48] due to the increase in Ca^2+^ ATPase of EPR and PM. NAADP has been found to release Ca^2+^ from lysosomal organelles through TPCs, and this, in turn, causes the release of Ca^2+^ from the ER store, which has possible morphological and functional links with acidic cell compartments in rat hepatocytes [18]. In the present article, we have shown that the combined influence of bafilomycin A1 and NAADP prevents the effect of NAADP on the activity of Ca^2+^ ATPases EPR in the subcellular fraction of rat liver (Figure 3). This demonstrates the relationship between acid storage and EPR in rat liver cells. Bafilomycin A1 most likely inhibited the H^+^ gradient at the acid store’s membranes, which is what drives the transit of Ca^2+^ ions, by blocking the acid store’s H^+^ pump. As a result, NAADP did not release calcium, which in turn did not alter the Ca^2+^ activity of the EPR pump. This confirms that NAADP-responsive receptors are localized in the endo-lysosomal store closely associated with the EPR. Apparently, there is some morphological contact between the EPR and the endo-lysosomal store [36,37], persisting even after obtaining the subcellular fraction by centrifugation. The combined effect of bafilomycin A1 and NAADP prevented the effect of NAADP on the activity of Ca^2+^ ATPases of PM; bafilomycin alone caused no change in the activity of this pump but prevented the effect of NAADP. This implies that the NAADP-induced increase in the activity of the Ca^2+^ ATPases of the PM is also dependent on a bafilomycin-sensitive store. Thus, our experiments in rat hepatocytes demonstrate that bafilomycin A1, targeting the endo-lysosomal system, that is, the NAADP-sensitive Ca^2+^ store, effectively modulates ATPase activity in membranes that are in close co-localization with the acidic store.

Initially, the experiments were performed on rat hepatocytes to investigate the effect of bafilomycin A1 on Ca^2+^ stores. We found that Ca^2+^ stores in rat hepatocytes decreased with the application of bafilomycin A1 (20 and 0.04 mM). Since it is known [52] that CTC is utilized to monitor Ca^2+^ signal from the EPR (pH 7.2) as well as from weakly acidic organelles presented by endosomes with pH between 6 and 6.4, it is likely that this drop (Figure 4A) represents a change in Ca^2+^ concentration primarily in the EPR and/or endosomes. Lysosomes have substantially lower pH values, typically between 4.5 and 5 [53]. This decrease in the stored calcium in rat hepatocytes by bafilomycin (20 and 0.04 μM) can be explained as follows (Figure 6): the direct binding of bafilomycin A1 to the H^+^ pump of the acid store membranes prevents the filling of these organelles with calcium because the driving force for its transport—the proton gradient—is disrupted, releasing calcium from the lysosomes, as shown for epithelial cell lines [54], which should create local areas of increased calcium concentration like “hot spots” [55,56,57], activating the EPR Ca^2+^ pump to fill the EPR, with possible subsequent release of Ca^2+^ from the EPR due to overload, and/or activating the EPR Ca^2+^ channels, possibly via Ca^2+^ induced Ca^2+^ release (CIRC). The interaction between various calcium stores, attributed to the Calcium-Induced Calcium Release (CIRC) mechanism, has also been documented by other researchers [37,58]. The lysosomal Ca^2+^ release, which was caused by bafilomycin A1, may also be amplified by the release of Ca^2+^ from EPR through the CICR. Thus, applying bafilomycin A1 (20 and 0.04 mM) caused decreased calcium content in EPR after calcium release from acidic stores (autophagosomes, late endosomes, and lysosomes). The effect of bafilomycin A1 (0.04 mM) is stronger than that of bafilomycin A1 (20 mM) because, at higher concentrations, bafilomycin A1 likely inhibits SERCA to pump Ca^2+^ into the ER, so no subsequent CICR occurs (Figure 4).

The absence of changes in stored Ca^2+^ under the action of bafilomycin A1 (0.001 mM) (see Figure 4A) could be explained by the fact that its action at this concentration is restricted only to lysosomes and does not affect other Ca^2+^ stores, such as EPR and therefore does not alter the fluorescence intensity of the Ca^2+^–CTC complex therein. Next, the effect of bafilomycin A1 (0.001 mM) on NAADP-induced calcium release in permeabilized rat hepatocytes was examined (Figure 4B) to determine the role of acidic stores (autophagosomes, late endosomes, and lysosomes) in this process. It is known that NAADP is able to release Ca^2+^ from acidic stores [28,59], which has been confirmed for various tissues [16,35], including rat hepatocyte microsomes [60], liver lysosomes [28], and rat hepatocytes [18]. Previously, it was shown that the effects of NAADP on stored Ca^2+^ depend on the concentration of buffers in the cell incubation medium of rat hepatocytes [18].

Thus, two different types of solutions were performed with the addition of an EGTA buffer (Figure 4) or without an EGTA buffer (Figure 4B). It was observed that the fluorescence intensity of the Ca^2+^–CTC complex in permeabilized hepatocytes did not change with simultaneous exposure to bafilomycin A1 (20 or 0.04 mM) and NAADP (7 μM) compared with the medium with bafilomycin A1 alone in a medium without EGTA–Ca^2+^ buffer (Figure 4B). This indicates that bafilomycin A1 (20 μM) effectively inhibits the effect of NAADP. In contrast, the simultaneous action of NAADP (7 μM) and a lower concentration of bafilomycin (0.001 mM) resulted in an increase in stored calcium, which we assume is due to its accumulation in the EPR. This is possible because, in the series described above, the incubation medium was used without EGTA–Ca^2+^ buffer. When used in different combinations to maintain free calcium at physiological levels, bafilomycin A1 (0.001 μM) did not alter the stored calcium content of rat hepatocytes (Figure 4), but it prevented the effects of NAADP. This is confirmation that the bafilomycin-sensitive Ca^2+^ store is also NAADP-sensitive, corresponding to the endo-lysosomal organelle of the cell. In addition, the effect of bafilomycin A1 on NAADP-induced changes in Ca^2+^ content in rat hepatocytes was found to be dependent on the presence of EGTA–Ca^2+^ buffer in the cell incubation medium. This may indirectly confirm that Ca^2+^ “hotspots” and/or CICR play an important role in the interplay between the lysosome and EPR in bafilomycin A1-induced Ca^2+^ release in rat hepatocytes, as hypothesized above.

Thus, the obtained results show promise for modulating ion transport across the membrane of cancer cells by affecting the “acid stores” (autophagosomes, late endosomes, and lysosomes), which could be used as a potential new approach for the treatment of colorectal cancer. Moreover, with the results, potential therapeutic targets for the treatment of cancer could be identified and contribute to the understanding of the mechanisms behind carcinogenesis and the function of NAADP-sensitive acid storage in these processes.

## 4. Materials and Methods

### 4.1. Ethical Standards and Characteristics of Patients

Under the authorization of the Bioethics Committee of the Biological Faculty of the National Ivan Franko University of Lviv, Protocol No. 11/10, 2022, all procedures involving animals were carried out in compliance with the “International Convention for Working with Animals”. The Institutional Review Board (Ethics Committee) of the Department of Therapy No. 1, Medical Diagnostics and Haematology and Transfusiology of the Faculty of Postgraduate Education, Danylo Halytsky Lviv National Medical University, approved the study using human samples on 2 September 2022, in accordance with the Declaration of Helsinki’s guidelines. A total of 20 patients with colorectal cancer (mean age 54.3 ± 1.7 years) were complexly studied, all of whom agreed to give colonic mucosa samples during endoscopy.

We analyzed cancer samples from 12 women and 8 men. Six women had colorectal cancer type I, and 6 had type II (Table 1). Among men, 5 patients had cancer of type I and 3 had type II. Neoplasms without nodes and metastases, when the cancer affected the submucosa, were included in the It type of cancer. The type II tumor invades through the muscularis propria into the subserosa with no nodes and no metastases.

Samples of the patients’ colonic mucosa were taken during endoscopy from both the cancer-affected and healthy (control) regions of their mucosa. Two samples—one of malignant tissue and the other of unaltered tissue—were taken from a single patient (control). The vehicle control of the study had the same parameters as “control” because we added to the control the same volume of solution without reagents. Before being a part of the initiative, every patient signed an informed permission form for diagnosis and tissue sample research.

### 4.2. Rat Hepatocytes Isolation

The “two-step” collagenase II type (Sigma, Burlington, MA, USA) perfusion method was used to generate hepatocytes, as previously mentioned [18]. Briefly, male wild-type rats (180–200 g) were anesthetized by inhalation of chloroform and then decapitated. The liver was isolated and digested using the “two-step” collagenase perfusion method as recommended for the preparation of isolated hepatocytes [61]. The liver is nonrecirculatingly perfused with Hank’s Balanced Salt Solution (HBSS), which is free of Ca^2+^ and Mg^2+^ and contains EGTA (0.1 mM) for 10 min. The liver is then briefly washed with EGTA-free HEPES-containing (10 mM) buffer at 37 °C for 2 min. After that, the blanched liver was perfused for ten to fifteen minutes with 0.01% collagenase in 1 mM HBSS that contained Ca^2+^. Following the collagenase treatment, the liver was removed, chopped, and mixed with HBSS and Ca^2+^. The resulting cell suspension was filtered using successive nylon mesh filters and centrifuged at 50× *g* for a duration of two minutes. Cells were resuspended in Ca^2+^-containing HBSS supplemented with 10% fetal bovine serum after the supernatant was removed. Trypan blue exclusion was used to determine the vitality of hepatocytes, and each experiment’s preparation contained about 90% viable cells. HBSS contained (mM): 137 NaCl, 5.4 KCl, 0.2 Na_2_HPO_4_, 0.4 KH_2_PO_4_, 0.4 MgSO_4_, 1.3 CaCl_2_, 4.1 Na_2_CO_3_, and 5.6 glucose. For Ca^2+^ and Mg^2+^ free HBSS, CaCl_2_ and MgSO_4_ were omitted from the medium.

### 4.3. Chlortetracycline Chemiluminescent Imaging as a Quantitative Measure of Stored Calcium in Rat Hepatocytes

Stored calcium was measured by chemiluminescence of the Ca^2+^ chlortetracycline complex (CTC). This Ca^2+^-sensitive indicator is used to monitor the stored Ca^2+^ concentration [52,53] in the lumen of the organelle. Prior to Ca^2+^ imaging, isolated hepatocytes were permeabilized in suspension with saponin (0.1 mg/mL) for 10 min in an intracellular solution. Following an intracellular solution wash, the cells were treated with the suitable drug for ten minutes. Permeabilized hepatocyte suspensions were treated for 10 min in intracellular solutions containing the necessary reagents prior to being loaded with chlortetracycline (CTC). After that, an intracellular solution was used to wash the permeabilized cell suspensions. 20 NaCl, 120 KCl, 1.13 MgCl_2_, 1.3 CaCl_2_, 10 HEPES, 5 μg/mL oligomycin, 1 μg/mL rotenone, and 2 ATP (pѝ 7.0) were the contents of the intracellular solution (mM). Every step of the experiment was carried out at 37 °C. To adjust the amount of free calcium present in the permeabilized rat hepatocytes’ incubation medium, we used EGTA–Ca^2+^ buffer with the following Ca^2+^ concentrations (maxchelator.stanford.edu/CaEGTA–TS.htm): [Ca^2+^]_cyt._ = 2.47 nM (EGTA 100 μM, CaCl_2_ 1 μM); [Ca^2+^]_cyt._ = 243 nM (EGTA 100 μM, CaCl_2_ 50 μM); [Ca^2+^]_cyt._ = 240 nM (EGTA 50 μM, CaCl_2_ 25 μM). Hepatocytes were loaded with 100 μM CTC and allowed to sit at room temperature for 20 min without light in order to measure the changes in internal Ca^2+^ concentrations. This Ca^2+^ sensitive chemiluminescent probe is widely used in biological systems [52,53] to monitor the free internal Ca^2+^ concentration in the lumen of organelles by the following mechanism: CTC permeates in neutral, uncomplexed form and internal and external CTC concentrations are equal at equilibrium. A tiny rise in chemiluminescence occurs within the lumen as a result of Ca^2+^ complexing with CTC. Additionally, this complex attaches itself to the membrane’s inner surface, greatly enhancing chemiluminescence. As a result, it is believed that CTC-Ca^2+^ chemiluminescent signals the Ca^2+^ concentration in intracellular organelles following depolarization of the mitochondria.

As a selective inhibitor of H^+^ pumps V-type, bafilomycin A1 (Sigma, USA) was introduced in concentrations of 20, 0.04, and 0.001 mM, and NAADP (Sigma, USA) in concentrations of 7 μM. Cells were plated on glass slides for Ca^2+^ measurements, and a LUMAM-I-1 luminous microscopy system (40× objective, NA 0.7) was used to examine individual cells. A light-emitting diode was used to stimulate the CTC at 380 nm, and emission fluorescence was measured between 505 and 520 nm. Each subset of studies involved the examination of a set of thirty cells within the field of view. The relative variations in Ca^2+^ in CTC chemiluminescence were standardized to a 100% control value. 

### 4.4. Assay of ATPase Activity in Subcellular Post-Mitochondrial Fraction of Rat Liver and Human Samples of Colon Mucosa

Isolation of the subcellular post-mitochondrial fraction was performed as previously described for rat liver [23,48,62] and for human colon mucosa and human colorectal samples [17], the determination of ATPase activity was based on the determination of the content of inorganic phosphorus by the spectrophotometric method. ATPase activity was estimated in the post-mitochondrial subcellular fraction obtained by differential centrifugation. In summary, tissue samples were homogenized at 0 °C to 2 °C for 10 min at 300 rpm using a glass-glass homogenizer. Supernatant 1 included mitochondria, whereas the homogenate was centrifuged at 3000× *g* for 10 min using a Jouan MR 1812 centrifuge (Saint-Herblain, France) to precipitate nuclei, large cell fragments, and undisturbed cells. Following another centrifugation of this supernatant for 10 min at 8500× *g* (0–2 °C), the mitochondrial fraction was sedimented. Supernatant 2, or the subcellular post-mitochondrial fraction, was separated after the mitochondria were sedimented, and it was utilized for the ATPase activity test. It was separated at 14,000× *g* for 20 min.

Using the Fiske-Subbarow method, which estimates the quantity of inorganic phosphorus (Pi) released during the ATP hydrolase reaction and expresses it in μmol Pi/mg protein per h, ATPase activity was evaluated spectrophotomically. The goal was to transfer the post-mitochondrial subcellular fraction, which was obtained using differential centrifugation, to an internal standard solution that contained the following components (in milligrams) at 37 °C: NaCI 50.0, KCl 100, Tris–ΝCl 20, MgCl_2_ 3, CaCl_2_ 0.01, and pѝ 7.0. The addition of 3 mM ATP (Sigma, Burlington, MA, USA) initiated the process. To measure their impact on ATPase activity, bafilomycin A1 (Sigma, USA) and NAADP (Sigma, USA) were added to the incubation suspension. The difference in Pi between the medium containing and excluding ouabain was used to indicate the activity of Na^+^/K^+^ ATPase (Sigma, USA). Instead, equivalent volumes of incubation media were present in the samples (devoid of ouabain). Baseline Mg^2+^ ATPase activity was assessed in an incubation medium with ouabain and EGTA but no CaCl_2_. It was decided to add thapsigargin (Sigma, USA) in order to suppress the Ca^2+^ ATP-dependent EPR. Additionally, the difference in Pi between the medium containing inhibitors (thapsigargin and ouabain) and the one without these substances was used to quantify the activity of Ca^2+^ ATP of EPR. The incubation medium was present in identical amounts in the samples (which did not contain thapsigargin or ouabain). Prior to being introduced to the incubation medium at a concentration of 1 μmol, thapsigargin and ouabain were dissolved in DMSO in a separate aliquot and then dissolved in an internal solution in another aliquot. Other samples (apart from thapsigargin and ouabain) had the same quantity of an incubation medium present. Micromoles of inorganic phosphorus, or 1 mg of protein every hour (μmol Pi/mg protein per hour), were used to express ATPase activity.

### 4.5. Specific ATPase Activity Calculation

The difference of inorganic phosphorus in the media with varying compositions was used to calculate the total ATPase activity of the post-mitochondrial fraction: (a) specific Na^+^/K^+^ ATPase activity was calculated as the difference of inorganic phosphorus content in the medium with or without ouabain (1 mM); (b) the difference between total Ca^2+^/Mg^2+^ and Na^+^/K^+^ ATPase activity was quantified; (c) the amount of SERCA (sarcoendoplasmic reticulum Ca^2+^ ATPase) that contributes to the overall Ca^2+^/Mg^2+^ ATPase activity was calculated using thapsigargin (d) in an incubation medium without ouabain and containing 1 mM EGTA. The incubation medium served as a control for the enzymatic hydrolysis of ATP in each experiment.

### 4.6. Statistical Analysis

The data in the text are expressed as mean ± SEM [63]. ANOVA test to estimate the significance of differences between experimental groups: *p* ˂ 0.05 was conducted with the statistical software using Origin Pro 2018 (www.originlab.com).

## 5. Conclusions

Bafilomycin A1 significantly raised the basal ATPase activity in both the normal mucosa and the subcellular portion of colon cancer tissue in patient colon samples. It also successfully boosted the Ca^2+^ ATPase of EPR activity. It was shown that bafilomycin A1 decreased the activity of Ca^2+^ ATPase of PM in human colon mucosal tissue samples and caused no changes in the activity of Ca^2+^ ATPase of PM in the subcellular fraction of human colon cancer tissue. It was shown that bafilomycin A1 decreased the activity of Na^+^/K^+^ ATPase in human colon mucosa tissue samples as well as in cancer samples. In rat hepatocytes, bafilomycin A1 effectively reduced calcium stores and blocked NAADP’s impact on them. In the subcellular fraction of rat liver, bafilomycin A1 decreased Na^+^/K^+^ ATPase activity and elevated baseline Mg^2+^ and EPR Ca^2+^ ATPase activities. When compared to the medium containing NAADP, the combination action of bafilomycin A1 and NAADP fully inhibits any change in Ca^2+^ ATPase and basal Mg^2+^ ATPase activities. The obtained results support the following statement: the EPR Ca^2+^ ATPases and the bafilomycin-sensitive storage are in close functional and physical contact. We explained the effect of bafilomycin A1 on these ATPase activities by acidification. Thus, the obtained results show promising targets for modulating ion transport across the membrane of cancer cells by affecting the “acidic stores” (autophagosomes, late endosomes, and lysosomes) as a possible new approach for the treatment of colorectal cancer.

## Figures and Tables

**Figure 1 ijms-25-01657-f001:**
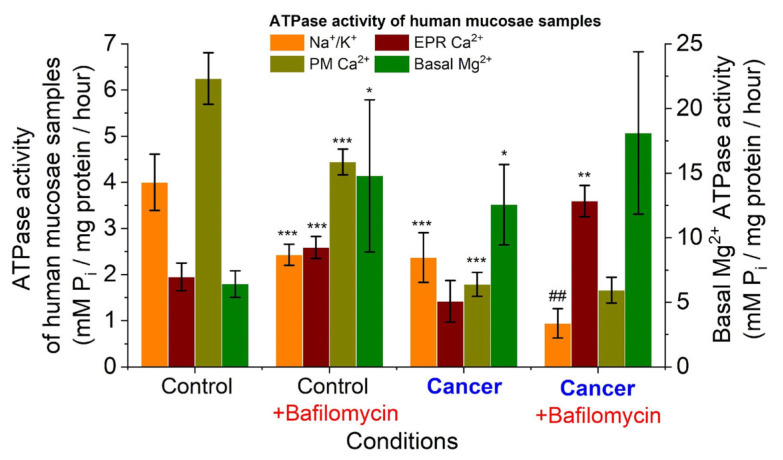
ATPase activity of subcellular fraction of colorectal cancer samples (cancer) and unchanged tissue (control): *** *p* ≤ 0.001 vs. control; ** *p* ≤ 0.01 vs. control; * *p* ≤ 0.05 vs. control, ## *p* ≤ 0.05 vs. cancer.

**Figure 2 ijms-25-01657-f002:**
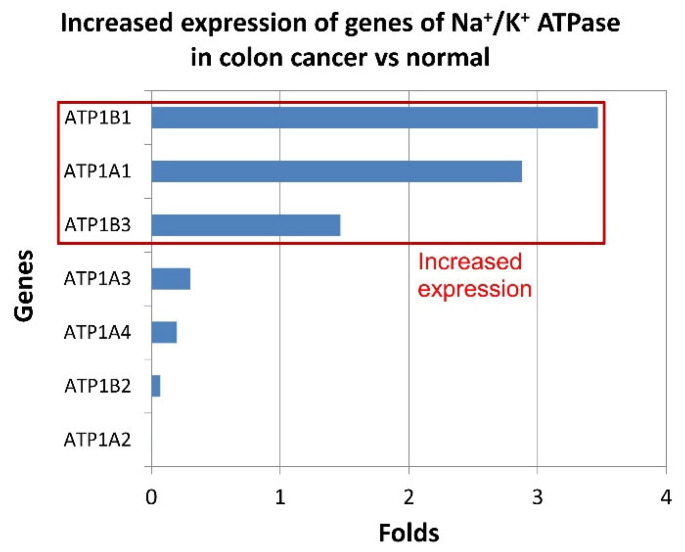
Increased expression of Na^+^/K^+^ ATPase’ genes subunits in colon cancer vs. normal (according to Gepia (http://gepia.cancer-pku.cn/detail.php?gene, accessed on 9 January 2024).

**Figure 3 ijms-25-01657-f003:**
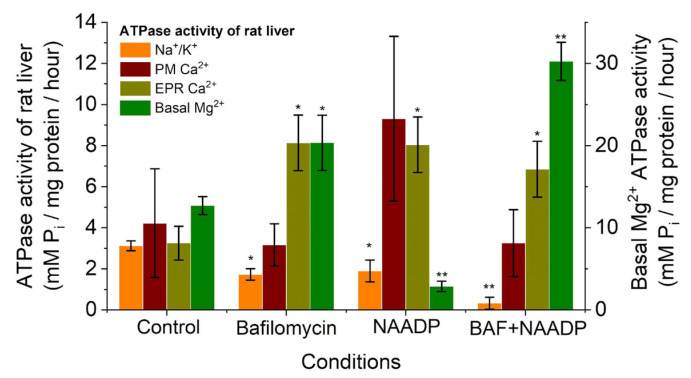
Simultaneous effect of bafilomycin A1 (0.001 mM) and NAADP (7 μM) on ATPase activity of rat liver post mitochondrial fraction: * *p* ≤ 0.05 vs. control; ** *p* ≤ 0.01 vs. control.

**Figure 4 ijms-25-01657-f004:**
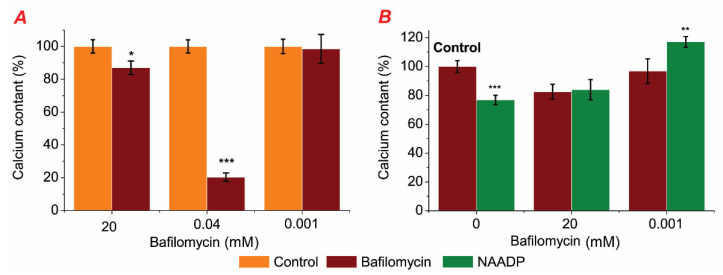
Bafilomycin (20; 0.04; 0.001 mM) addition effect in the medium without EGTA–Ca^2+^ buffers on calcium content in permeabilized rat hepatocytes (**A**) and simultaneous effect of bafilomycin (20, 0.001 mM) and NAADP (7 μM) on calcium content in permeabilized rat hepatocytes (**B**): * *p* ≤ 0.05 vs. control, ** *p* ≤ 0.01 vs. NAADP, *** *p* ≤ 0.001 vs. control.

**Figure 5 ijms-25-01657-f005:**
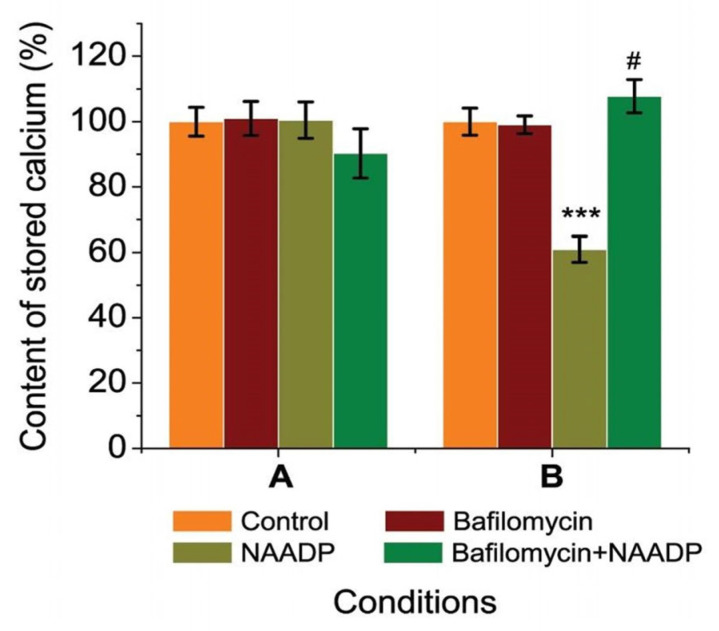
Simultaneous effect of bafilomycin (0.001 mM) and NAADP (7 μM) on calcium content (%) in permeabilized rat hepatocytes in the medium with different EGTA–Ca^2+^ buffers: [Ca^2+^]_cyt_. = 243 nM (EGTA: 100 μM, CaCl_2_: 50 μM) (A); [Ca^2+^]_cyt_. = 240 nM (EGTA: 50 μM, CaCl_2_: 25 μM) (B). *** *p* ≤ 0.001 vs. control; # *p* ≤ 0.01 vs. NAADP.

**Figure 6 ijms-25-01657-f006:**
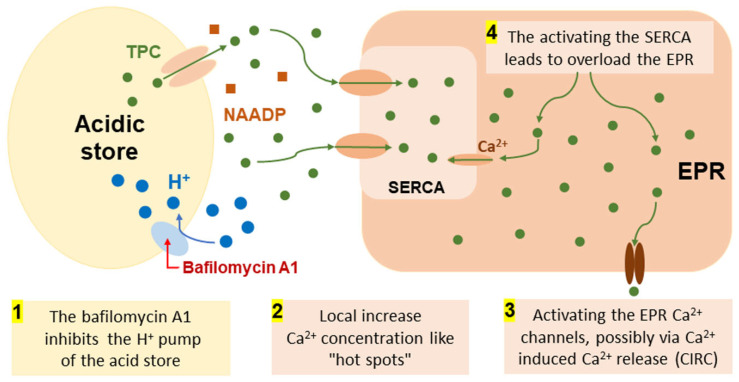
The hypothetical scheme of the molecular mechanisms of bafilomycin A1 effect on acid Ca^2+^ store and EPR: the bafilomycin A1 inhibits the H^+^ pumps of acid store (1), this creates a local areas of increased calcium concentration like “hot spots” close to membranes of EPR (2), Activating the SERCA to fill the EPR by Ca^2+^ (3), this leads to overload the EPR and next activating the EPR Ca^2+^ channels (4), possibly via Ca^2+^ induced Ca^2+^ release (CIRC). Explanation: green dots and arrows - movement of Ca^2+^ ion; blue dots and arrows - movement of H^+^ protons; the red arrow indicates the inhibition of the H^+^-pump by bafilomycin A1; brown squares are NAADP molecules.

**Table 1 ijms-25-01657-t001:** The characteristics of patients, types, and stages of colorectal cancer.

Number of pts.	Age of pts.	Sex	Stage	Definition
1	53	female	I	The tumor invades the submucosa—no nodes, no metastases
2	52	female	I	The tumor invades the submucosa—no nodes, no metastases
3	76	female	II	The tumor invades through the muscularis propria into the subserosa—no nodes, no metastases
4	53	female	II	The tumor invades through the muscularis propria into the subserosa—no nodes, no metastases
5	52	female	I	The tumor invades the submucosa—no nodes, no metastases
6	50	female	II	The tumor invades through the muscularis propria into the subserosa—no nodes, no metastases
7	74	female	I	The tumor invades the submucosa—no nodes, no metastases
8	51	female	I	The tumor invades the submucosa—no nodes, no metastases
9	51	female	I	The tumor invades the submucosa—no nodes, no metastases
10	48	female	II	The tumor invades through the muscularis propria into the subserosa—no nodes, no metastases
11	45	female	II	The tumor invades through the muscularis propria into the subserosa—no nodes, no metastases
12	56	female	II	The tumor invades through the muscularis propria into the subserosa—no nodes, no metastases
13	55	male	I	The tumor invades the submucosa—no nodes, no metastases
14	45	male	I	The tumor invades the submucosa—no nodes, no metastases
15	56	male	I	The tumor invades the submucosa—no nodes, no metastases
16	55	male	II	The tumor invades through the muscularis propria into the subserosa—no nodes, no metastases
17	50	male	II	The tumor invades through the muscularis propria into the subserosa—no nodes, no metastases
18	54	male	II	The tumor invades through the muscularis propria into the subserosa—no nodes, no metastases
19	50	male	I	The tumor invades the submucosa—no nodes, no metastases
20	60	male	I	The tumor invades the submucosa—no nodes, no metastases
M	**54.3**			
m	**1.7**			

## Data Availability

The datasets generated and/or analyzed during the current study are not publicly available due to privacy but are available from the corresponding author upon reasonable request.

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
