# Peer review of "Bafilomycin A1 Molecular Effect on ATPase Activity of Subcellular Fraction of Human Colorectal Cancer and Rat Liver"

_ijms, 2024, doi:10.3390/ijms25031657_

Round 1
Reviewer 1 Report (New Reviewer)
Comments and Suggestions for Authors
The manuscript prepared by Ivan’s group illustrated the impact of bafilomycin A1 on Ca2+ content, NAADP-induced Ca2+ release, and ATPase activity in both rat and human cancer samples. The findings of this study are vital which explored the potential of the usage of bafilomycin A1 for the treatment of cancer. Overall, this manuscript is well-written, no major concerns were found.
Minor: Please provide the vehicle control of the study.
Author Response
Dear Reviewer,
Thank you very much for your time and positive evaluation of our manuscript. We would especially thank you for your positive recommendations and comments. We have improved our manuscript according to your important comments (lines 514,515):
The vehicle control of the study had the same parameters as “control”, because we add to the control the same volume of solution without reagent.
All changes to the manuscript have been highlighted.
Reviewer 2 Report (New Reviewer)
Comments and Suggestions for Authors
In this manuscript, the authors investigated the effects of Bafilomycin on the activity of Na+/K+ ATPase and Mg2+ ATPase in rat liver cells, human colon, and human colon cancer samples, as well as its combined effect with NAADP. However, some revisions and additions are needed.
1. It is suggested that the authors investigate the specific molecular effects of Bafilomycin A1, such as how it affects Na+/K+ ATPase and Mg2+ ATPase activity, upstream and/or downstream.
2. In Figure 4A, when we compared the 20 mM group and the 0.04 mM group of Bafilomycin, less Bafilomycin decreased more Ca2+. Is there any possible explanation for this?
3. The manuscript has some strikethroughs or corrections that need to be further verified and corrected.
4. In line 260, it should be Figure 5B, not Figure 3B.
Author Response
Dear Reviewer,
Thank you very much for your time and positive evaluation of our manuscript. We would especially thank you for your positive recommendations and comments. We have improved our manuscript according to your important comments.
Our responses are below:
-
It is suggested that the authors investigate the specific molecular effects of Bafilomycin A1, such as how it affects Na+/K+ ATPase and Mg2+ ATPase activity, upstream and/or downstream.
This is downstream effect of bafilomycin A1 on these pumps, which is as consequence of effect of bafilomycin A1 on H+ pump of the acid store membranes. Binding of bafilomycin A1 prevents the filling of these organelles with calcium because the driving force for its transport – the proton gradient – is disrupted. In the same time it created local higer concetration of H+ and Ca2+, which affect on ion transport by Na+/K+-pump and Mg2+ ATPase activity.
-
In Figure 4A, when we compared the 20 mM group and the 0.04 mM group of Bafilomycin, less Bafilomycin decreased more Ca2+. Is there any possible explanation for this?
Thank you for this important question.
We explain this observation due to Ca2+-dependent cross-talk between acidic store and ER Ca2+-store: released from acidic store calcium is immediatlly uptaked by SERCA, which pump it into ER. The lysosomal Ca2+ release, which was caused due to bafilomycin A1 may be amplified by the release Ca2+ from EPR through the CICR too. Thus, applying bafilomycin A1 (20 and 0.04 mM) caused decreased calcium content in EPR after calcium release from acidic stores (autophagosomes, late endosomes, and lysosomes).
The effect of bafilomycin A1 (0.04 mM) is stronger than that of bafilomycin A1 (20 mM) because at higher concentrations, bafilomycin A1 likely inhibits SERCA to pump Ca2+ into the ER, so no subsequent CICR occurs.
We have added these explanation into lines 446-448
-
The manuscript has some strikethroughs or corrections that need to be further verified and corrected.
Thank you. The strikethroughs will be removed during final proof-reading in case you should hopefully accept our manuscript for publication.
4. In line 260, it should be Figure 5B, not Figure 3B.
Thank you very much. We have improved it in the manuscript
Round 2
Reviewer 2 Report (New Reviewer)
Comments and Suggestions for Authors
I recommend this work to be published in this journal.
This manuscript is a resubmission of an earlier submission. The following is a list of the peer review reports and author responses from that submission.
Round 1
Reviewer 1 Report
Comments and Suggestions for Authors
The authors did not address my concern. I am not satisfy with their replies and additional experiments they have provided to support their hypothesis. I aggree with another reviewer, this article should not be published in IJMS.
Comments on the Quality of English LanguageNA
Author Response
Reviewer 1
The authors did not address my concern. I am not satisfy with their replies and additional experiments they have provided to support their hypothesis. I agree with another reviewer, this article should not be published in IJMS.
Dear reviewer 1,
Another reviewer evaluated our manuscript with positive following comments “Given the increasing burden of colorectal cancer, this is a good topic to investigate and the authors have mostly carried out the study well. However, there are some issues that need to be addressed and more data analysis is required prior to acceptance. The English used is good and can be easily understood by readers. Minor spell check should be done.
We are so so sorry that this reviewer is not satisfied our improved and is so negative.
Thank you for your time and evaluation of our manuscript.
Best wishes,
Dr. Ivan Kushkevych, corresponding author and my co-authors.
Reviewer 2 Report
Comments and Suggestions for Authors
The article by Bychkova and colleagues investigates the effect of bafilomycin A1 on colon cancer and rat liver hepatocytes. Given the increasing burden of colorectal cancer, this is a good topic to investigate and the authors have mostly carried out the study well. However, there are some issues that need to be addressed and more data analysis is required prior to acceptance.
1. Figure 4, baf1 group needs to be of the same color for consistency.
2. Could the authors provide a table describing the statistics of the patients and the disease stage of the cancer?
3. The authors have stated that they wanted to check the effect of Baf1 on Ca2++ reserves in hepatocytes. What was the justification for using rat hepatocytes? Could the authors not perform all the experiments from the control adjacent tissue or from healthy rat colons?
4. Did the authors observe a difference between the EPR Ca2++ levels after Baf1 treatment in adjacent control colon tissue vs rat hepatocytes?
5. In the schematic, the author’s state that Baf1 inhibits H+ pumps, however, the authors have used red sharp arrow to depict that. Either a blunt arrow should be used or the sharp arrows need to be annotated, for e.g. red arrow=inhibition, green arrow=activation.
Comments on the Quality of English LanguageThe English used is good and can be easily understood by readers. Minor spell check should be done.
Author Response
Reviewer 2
The article by Bychkova and colleagues investigates the effect of bafilomycin A1 on colon cancer and rat liver hepatocytes. Given the increasing burden of colorectal cancer, this is a good topic to investigate and the authors have mostly carried out the study well. However, there are some issues that need to be addressed and more data analysis is required prior to acceptance.
Dear Reviewer 2,
Thank you very much for your time and positive evaluation of our manuscript. We would especially thank you for your positive recommendations and comments. We have improved our manuscript according to your important comments.
Best wishes,
Ivan Kushkevych, corresponding author and my co-authors.
Our responses are below:
-
Figure 4, baf1 group needs to be of the same color for consistency.
Thank you. We agree with comment
-
Could the authors provide a table describing the statistics of the patients and the disease stage of the cancer?
We analysed cancer samples from 12 women and 8 men. 6 women had colorectal cancer type I and 6 - type II. Among men: 5 patients had cancer of type I and 3 - type II. Neoplasms without nodes and metastases, when the cancer affected the submucosa, were included in the It type of cancer. The II type - tumour invades though muscularis propria into subserosa, no nodes no metastases
Table 1. The characteristics of patients, typs and stages of colorectal cancer.
|
# of pts. |
Age of pts. |
Sex |
Stage |
Definition |
|
1 |
53 |
female |
I |
tumour invade submucosa, no nodes, no metastases |
|
2 |
52 |
female |
I |
tumour invade submucosa, no nodes, no metastases |
|
3 |
76 |
female |
II |
tumour invades though muscularis propria into subserosa, no nodes no metastases |
|
4 |
53 |
female |
II |
tumour invades though muscularis propria into subserosa, no nodes no metastases |
|
5 |
52 |
female |
I |
tumour invade submucosa, no nodes, no metastases |
|
6 |
50 |
female |
II |
tumour invades though muscularis propria into subserosa, no nodes no metastases |
|
7 |
74 |
female |
I |
tumour invade submucosa, no nodes, no metastases |
|
8 |
51 |
female |
I |
tumour invade submucosa, no nodes, no metastases |
|
9 |
51 |
female |
I |
tumour invade submucosa, no nodes, no metastases |
|
10 |
48 |
female |
II |
tumour invades though muscularis propria into subserosa, no nodes no metastases |
|
11 |
45 |
female |
II |
tumour invades though muscularis propria into subserosa, no nodes no metastases |
|
12 |
56 |
female |
II |
tumour invades though muscularis propria into subserosa, no nodes no metastases |
|
13 |
55 |
male |
I |
tumour invade submucosa, no nodes, no metastases |
|
14 |
45 |
male |
I |
tumour invade submucosa, no nodes, no metastases |
|
15 |
56 |
male |
I |
tumour invade submucosa, no nodes, no metastases |
|
16 |
55 |
male |
II |
tumour invades though muscularis propria into subserosa, no nodes no metastases |
|
17 |
50 |
male |
II |
tumour invades though muscularis propria into subserosa, no nodes no metastases |
|
18 |
54 |
male |
II |
tumour invades though muscularis propria into subserosa, no nodes no metastases |
|
19 |
50 |
male |
I |
tumour invade submucosa, no nodes, no metastases |
|
20 |
60 |
male |
I |
tumour invade submucosa, no nodes, no metastases |
|
M |
54,3 |
|
|
|
|
m |
1,7 |
|
|
|
-
The authors have stated that they wanted to check the effect of Baf1 on Ca2++ reserves in hepatocytes. What was the justification for using rat hepatocytes? Could the authors not perform all the experiments from the control adjacent tissue or from healthy rat colons?
We used rat hepatocytes in a completely separate series of studies with the aim of finding out how Baf affects Ca2+-store in whole intact cells. Isolated hepatocytes are an excellent model object of whole cells with preserved intracellular connections, which makes it possible to determine the molecular effects of Baf, in particular, on the stored calcium. This technique is routinely used in our laboratory and there are no problems with obtaining viable cells for experiments. While studying the effects of Baf on isolated colorectal cancer cells from patients or adjacent healthy tissues requires isolation of such cells on a case-by-case basis and such experiments may not be reproducible from patient to patient. Therefore, we worked on the subcellular fraction obtained from colorectal cancer patients and from adjacent intact tissues. The subcellular fraction was obtained as a result of homogenization of the tissue and subsequent centrifugation with sedimentation of mitochondria.
-
Did the authors observe a difference between the EPR Ca2++ levels after Baf1 treatment in adjacent control colon tissue vs rat hepatocytes?
We measured the ATPase activity in all the investigated tissues and found the following similar effects: Baf stimulates Ca2+-ATPase EPR both in the liver and in unchanged adjacent tissue of the human colon, and it also inhibits the Na+-K-ATPase in all the investigated tissues. However, the effects of Baf in cancer were significantly more powerful than in healthy ones.
-
In the schematic, the author’s state that Baf1 inhibits H+ pumps, however, the authors have used red sharp arrow to depict that. Either a blunt arrow should be used or the sharp arrows need to be annotated, for e.g. red arrow=inhibition, green arrow=activation.
Thank you. We agree with comment
Comments on the Quality of English Language
The English used is good and can be easily understood by readers. Minor spell check should be done.
Thank you very much! We improved the manuscript.
Best wishes,
Ivan Kushkevych, corresponding author and my co-authors.
Round 2
Reviewer 1 Report
Comments and Suggestions for Authors
The authors did not address my concern. I am not satisfy with their replies.